# Anti-LRP5/6 VHHs promote differentiation of Wnt-hypersensitive intestinal stem cells

Nicola Fenderico [1], Revina C. van Scherpenzeel[2], Michael Goldflam[3,6], Davide Proverbio[4,7], Ingrid Jordens[1], Tomica Kralj[1], Sarah Stryeck[5], Tarek Z. Bass[4], Guy Hermans[3], Christopher Ullman[3,8], Teodor Aastrup[4], Piet Gros [2] & Madelon M. Maurice [1]

Wnt-induced β-catenin-mediated transcription is a driving force for stem cell self-renewal during adult tissue homeostasis. Enhanced Wnt receptor expression due to mutational inactivation of the ubiquitin ligases *RNF43/ZNRF3* recently emerged as a leading cause for cancer development. Consequently, targeting canonical Wnt receptors such as LRP5/6 holds great promise for treatment of such cancer subsets. Here, we employ CIS display technology to identify single-domain antibody fragments (VHH) that bind the LRP6 P3E3P4E4 region with nanomolar affinity and strongly inhibit Wnt3/3a-induced β-catenin-mediated transcription in cells, while leaving Wnt1 responses unaffected. Structural analysis reveal that individual VHHs variably employ divergent antigen-binding regions to bind a similar surface in the third β-propeller of LRP5/6, sterically interfering with Wnt3/3a binding. Importantly, anti-LRP5/6 VHHs block the growth of Wnt-hypersensitive *Rnf43/Znrf3*-mutant intestinal organoids through stem cell exhaustion and collective terminal differentiation. Thus, VHH-mediated targeting of LRP5/6 provides a promising differentiation-inducing strategy for treatment of Wnt-hypersensitive tumors.

[1] Oncode Institute and Department of Cell Biology, Center for Molecular Medicine, University Medical Center Utrecht, 3584 CX Utrecht, The Netherlands. [2] Crystal and Structural Chemistry, Department of Chemistry, Bijvoet Center for Biomolecular Research, Utrecht University, Utrecht 3584 CH, The Netherlands. [3] Isogenica Ltd., Chesterford Research Park, CB10 1XL Little Chesterford, Essex, UK. [4] Attana AB, SE-11419 Stockholm, Sweden. [5] Gottfried Schatz Research Center for Cell Signaling, Metabolism and Aging, Molecular Biology and Biochemistry, Medical University of Graz, 8010 Graz, Austria. [6] Present address: Pepscan Therapeutics, 8243 RC Lelystad, The Netherlands. [7] Present address: NovAliX, Illkirch 67400, France. [8] Present address: Paratopix Ltd., Bishop's Stortford CM23 5JD, UK. These authors contributed equally: Nicola Fenderico, Revina C. van Scherpenzeel, Michael Goldflam. Correspondence and requests for materials should be addressed to M.M.M. (email: M.M.Maurice@umcutrecht.nl)

Wnt/β-catenin signaling is a critical driver of stem cell self-renewal and cell fate specification during development and adult tissue homeostasis[1]. Inappropriate activation of Wnt/β-catenin signaling due to mutations is a frequent event during the onset and progression of human cancer[2]. In healthy cells, Wnt/β-catenin signaling is initiated by binding of Wnt proteins to members of the Frizzled (FZD) receptor family and their co-receptors low-density lipoprotein receptor-related protein 5 or 6 (LRP5/6)[3]. Wnt-mediated receptor activation inhibits proteolytic turnover of β-catenin by the destruction complex, a multiprotein assembly composed of the scaffold proteins AXIN and Adenomatous Polyposis Coli (APC) and the kinases GSK3β and CK1[4]. As a consequence, β-catenin accumulates, migrates to the nucleus and drives transcription of Wnt target genes[1]. The duration and amplitude of Wnt-mediated cellular responses are balanced by negative regulatory feedback components such as the membrane-bound PA-RING ubiquitin ligases RNF43 and ZNRF3 that mediate internalization and lysosomal degradation of FZD receptors[5,6]. Within the stem cell niche, however, Wnt signals are locally kept elevated by secreted R-spondin (Rspo) proteins that bind Lgr4/5/6 receptors to capture and neutralize RNF43/ZNRF3 activity[7,8].

Mutations in β-catenin destruction complex components, such as APC and β-catenin, are well-known causal events in human colorectal cancer (CRC)[9,10]. More recently, inappropriately increased Wnt receptor activity due to genetic alterations in *RNF43, ZNRF3, RSPO2/3*[5,6,11–13] and *LRP6*[14] emerged as an alternative pathway for CRC development[10]. Such tumors display hypersensitivity to Wnt/Rspo stimulation, making them a high-priority target for the development of ligand or receptor blocking therapeutic interventions[15]. Indeed, inhibitors of Porcupine, an essential acyltransferase in Wnt secretion, are currently in clinical trials for treatment of Wnt-dependent tumors[16]. Due to their broad Wnt specificity, Porcupine inhibitors block both β-catenin-dependent and -independent Wnt signaling pathways and thus might cause major side effects, as exemplified by treatment-induced defects in bone turnover[17]. Of note, depletion of a single Wnt, Wnt3, entirely prevented formation of intestinal adenomas in double knock-out *Rnf43/Znrf3*-mutant (*R/Z* dKO) epithelia in mice, suggesting that more targeted approaches hold potential to eradicate Wnt-dependent tumors while diminishing side effects[15].

A key mediator of β-catenin-dependent Wnt signaling is the type I single-pass co-receptor LRP6[18,19]. The extracellular region of LRP6 comprises four YWTD-β-propeller-EGF domain modules (P1E1, P2E2, P3E3 and P4E4) and an LDLR-repeat domain preceding its transmembrane helix. The β-propeller-EGF modules harbor two independent Wnt binding sites. The first site, located within the N-terminal P1E1P2E2 domains, binds Wnt1, Wnt2, Wnt2b, Wnt6, Wnt8a, Wnt9a, Wnt9b and Wnt10b (site 1); while the second site, located within P3E3P4E4, binds Wnt3 and Wnt3a (site 2)[20–23]. The structural basis for this distinction in Wnt binding to LRP6 is not known. The activation of LRP6 in vivo is firmly controlled by extracellular antagonists such as DKK and SOST[24,25] that block Wnt binding and enhance receptor internalization[23,26–28]. In human cancer, epigenetic silencing of *DKK* is frequently observed, providing an additional route to inappropriately elevate Wnt-mediated signaling in cancer cells[29].

Domain-dependent Wnt binding to the LRP6 receptor offers an opportunity to selectively block certain classes of Wnts, while leaving other Wnt routes unaffected. The central role of LRP6 in Wnt/β-catenin signal relay in several cancer subsets has instigated the development of monoclonal antibodies (mAb) that interfere with Wnt binding and block receptor-dependent pathway activation[21,28,30–33]. Unexpectedly, however, mAb-mediated inhibition of Wnt binding to LRP6 site 1 strongly potentiated cellular responses to Wnts binding to site 2 and vice versa, likely due to mAb-mediated LRP6 dimerization[21,30]. These Wnt-enhancing properties complicate the application of LRP6-targeting mAbs in vivo, in a pathophysiological context.

Here, we screened a fully synthetic, highly diverse single-domain antibody fragment (VHH) library using CIS display technology[34,35]. Using functional assays, we selected three highly potent VHHs that bind LRP6 with nanomolar affinity and efficiently block Wnt3/3a-dependent β-catenin signaling. Structural analysis revealed that these VHHs all bind a surface of the third propeller domain of LRP6 that is likely involved in Wnt3 binding. Moreover, treatment with anti-LRP6 VHHs induces strong growth inhibition of Wnt-hypersensitive intestinal organoids by driving collective terminal differentiation. Thus, we identify a highly potent set of VHHs that target Wnt-hypersensitive tumors.

## Results

**Selection of anti-LRP6 VHHs.** We performed CIS display-selections on a library encoding >$10^{13}$ VHHs to isolate VHHs that bind the LRP6 Wnt3-binding domain[35–37]. To this end, recombinant human LRP6 β-propeller-EGF modules P3E3P4E4 (residues UNIPROT 629–1244) were secreted from human embryonic kidney (HEK) 293 cells (Fig. 1a). Purified LRP6$^{P3E3P4E4}$ showed a monodisperse peak after size-exclusion chromatography (SEC) and a single band on reducing SDS-PAGE (Supplementary Fig. 1). Selecting the library with LRP6$^{P3E3P4E4}$ and subsequent characterization of binding clones yielded 33 unique VHH clones. The vast majority of purified LRP6-binding VHHs substantially inhibited Wnt3a-mediated responses in HEK293T cells that overexpressed LRP6, as revealed by a luciferase-based Wnt reporter assay (TopFlash) (Fig. 1b). Moreover, endogenous Wnt3a-mediated pathway activation was reduced to <10% by half of the VHHs at 10 μM (Fig. 1c).

Next, we tested the most potent VHHs for inhibition of overexpressed and endogenous LRP6-dependent Wnt3a responses in a dose-dependent manner using 12.5, 2.5, 0.5 and 0.1 μM of each VHH. A VHH targeting an irrelevant antigen (human CD3) served as a negative control. Clear dose–response effects were observed for some VHHs, while others remained inhibitory at all doses tested (Fig. 2a, b). Next, we determined binding affinities for the three most potent VHH candidates (L-P2-B10, L-P2-D07 and L-P2-H07). Measurements of VHH-LRP6$^{P3E3P4E4}$ interactions in vitro by isothermal titration calorimetry (ITC) revealed low nanomolar range binding affinities (<40 nM) and the formation of a 1:1 complex with LRP6$^{P3E3P4E4}$ for each of the tested VHH (Fig. 2c). Thermodynamic parameters ($-T\Delta S$) however indicated that the binding mode of L-P2-H07 differs from that of L-P2-B10 and L-P2-D07. Differences are most likely due to variations in the number of entropy- (i.e., hydrophobic interactions) and enthalpy-driven reactions (hydrogen-bonding and van der Waals interactions). To validate these findings, L-P2-B10, L-P2-D07 and L-P2-H07 were taken forward for characterization of LRP6 binding kinetics using quartz crystal microbalance (QCM) analysis[38,39]. LRP6$^{P3E3P4E4}$ was immobilized on an Attana sensor chip followed by kinetic titration of VHH injections[40]. All three VHHs displayed nanomolar range binding affinities for LRP6$^{P3E3P4E4}$, characterized by relatively fast association rates ($k_{on}$: $10^5$/M/s) and mid-range dissociation rates ($k_{off}$: $10^{-4}$/s) (Supplementary Fig. 2).

**Structural basis of VHH-mediated LRP6 inhibition.** Based on functional data and binding kinetics, we investigated structural aspects of VHH-mediated LRP6 inhibition. Analytical SEC confirmed binding of all three VHHs to LRP6$^{P3E3P4E4}$ in solution (Supplementary Fig. 3). X-ray diffraction data were collected from

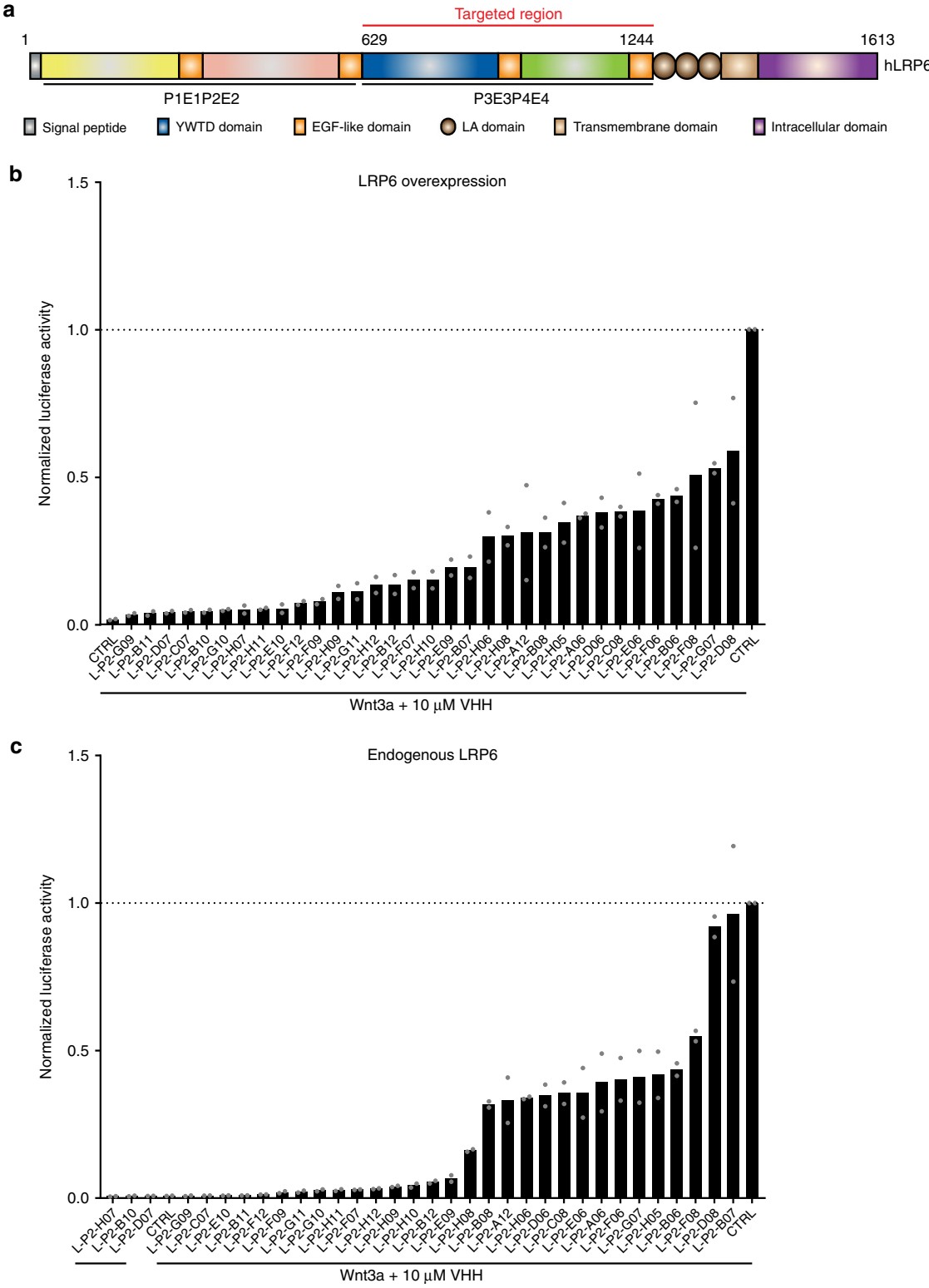

**Fig. 1** VHHs targeting LRP6$^{P3E3P4E4}$ block cellular responses to Wnt3a. **a** Schematic representation of LRP6. The P3E3P4E4 module of the extracellular domain was used to generate anti-LRP6 VHHs. Coloring scheme: LRP6$^{P1E1}$; yellow/orange, LRP6$^{P2E2}$; pink/orange, LRP6$^{P3E3}$; blue/orange and LRP6$^{P4E4}$; green/orange. LA domains are shown in brown. **b** Wnt luciferase reporter assay performed in LRP6-overexpressing HEK293T cells stimulated with Wnt3a-conditioned medium and treated with 10 μM of the indicated anti-LRP6$^{P3E3P4E4}$ VHHs. **c** Wnt luciferase reporter assay performed in HEK293T cells stimulated with Wnt3a-conditioned medium and treated with 10 μM of the indicated anti-LRP6$^{P3E3P4E4}$ VHHs. Graphs show average (bars) and range (dots) of luciferase activity in duplicate cell cultures transfected in parallel

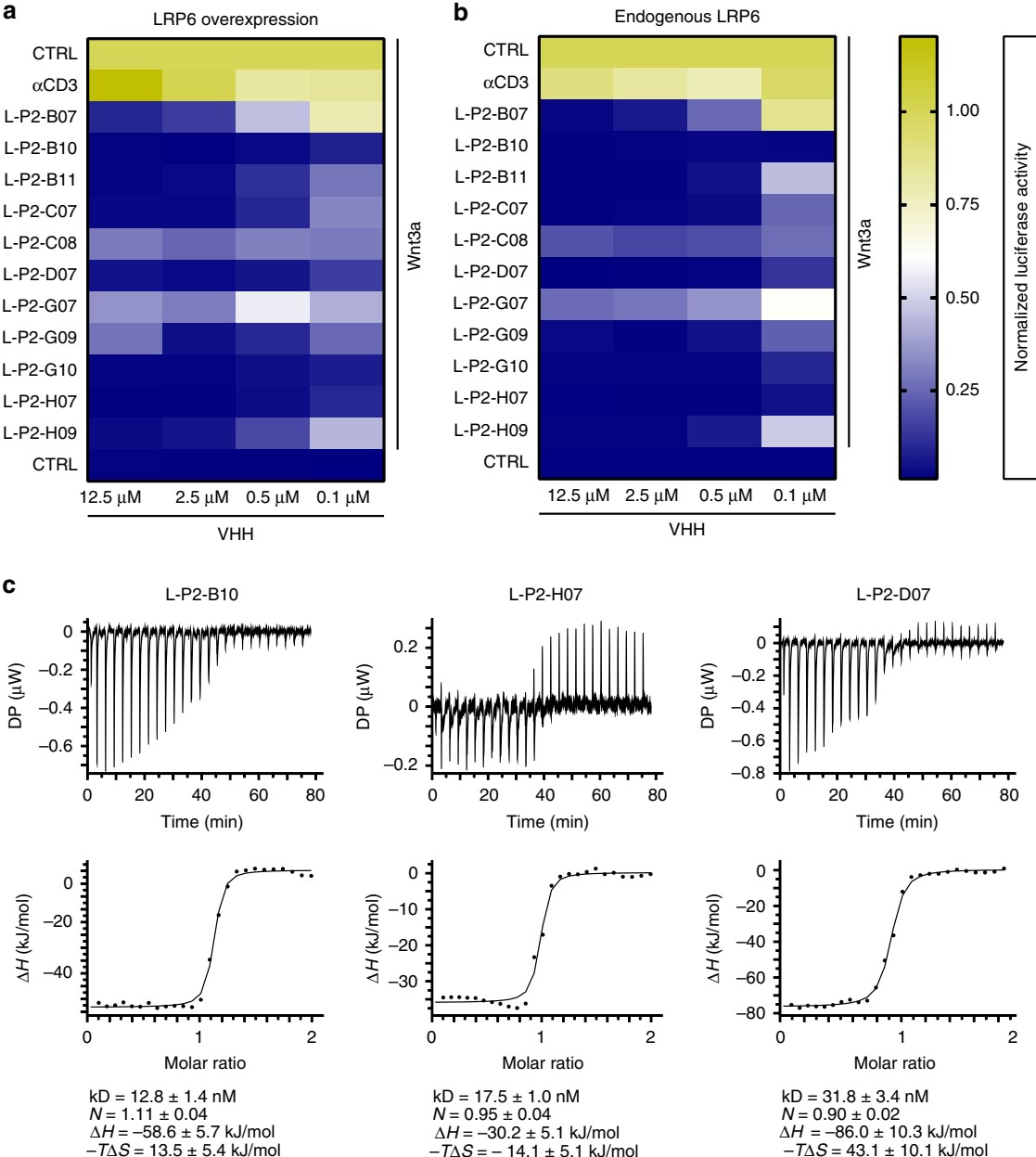

**Fig. 2** Anti-LRP6P3E3P4E4 VHHs show nM affinities. **a, b** Luciferase activities normalized to unstimulated control are represented as a heat map. Anti-LRP6P3E3P4E4 VHHs cause a dose-dependent decrease of Wnt signaling in both HEK293T cells overexpressing LRP6 (**a**) or untransfected HEK293T cells (**b**). **c** Binding affinities of anti-LRP6P3E3P4E4 VHHs as revealed by ITC measurements. One representative experimental titration curve of each VHH binding to LRP6P3E3P4E4 is shown. DP: differential power, kD: dissociation constant, N: stoichiometry, ΔH: delta enthalpy, −TΔS: temperature delta entropy

crystals of LRP6P3E3P4E4 in complex with L-P2-B10, L-P2-D07 and L-P2-H07 up to 2.6, 2.9 and 3.4 Å resolution, respectively (Table 1); however, data from the inter-grown crystals of LRP6P3E3P4E4-L-P2-H07 were of poor quality indicated by very high mosaicity, which allowed for initial structure solution by molecular replacement (MR), but not for detailed structure refinement. The three VHHs bind to a common site on the P3 domain of LRP6, without changing the LRP6P3E3P4E4 arrangement[26,27,41] (Fig. 3a, Supplementary Fig. 4).

L-P2-B10 and L-P2-D07 share the same epitope on LRP6P3, burying a total surface are of 1655 Å² and 1605 Å², respectively. At the center of the LRP6 blades residues Ile681, Trp767, Phe836, Trp850, Tyr875 and Met877 provide intermolecular hydrophobic contacts (Fig. 3b, c). In addition, both L-P2-B10 and L-P2-D07 interact with the LRP6P3 barrel through two clusters of polar and

electrostatic side-chain interactions, involving ten residues of LRP6P3. On one side of the barrel, Glu663 and Glu708 are forming salt bridges, while a second array of side-chain interactions occur at blades 5 and 6, with a salt bridge formed by Arg792 and Arg853 (Fig. 3b, c). Even though the two VHHs share the same epitope on LRP6, the paratopes of both VHHs are different. The CDR3 of L-P2-B10 is six residues longer than that of L-P2-D07 (Supplementary Fig. 5a), which explains a difference in binding orientation of the two VHHs (Supplementary Fig. 5b). The longer CDR3 loop of L-P2-B10 folds back over the VHH framework region, whereas the CDR3 loop of L-P2-D07 is directed towards the barrel. Additionally, the distribution of charged residues on the paratope comprising three CDRs is different. Markedly, the L-P2-B10 and L-P2-D07 epitopes strongly overlap with the binding site for DKK1_C, a natural

**Table 1 Crystallographic and structure refinement statistics**

| | LRP6 P3E3P4E4: L-P2-B10 | LRP6 P3E3P4E4: L-P2-D07 |
|---|---|---|
| *Data collection* | ESRF-ID23-2 | ESRF-ID29 |
| Space group | $P\,6_5$ | $P\,2_1 2_1 2_1$ |
| Cell dimensions | | |
| $a, b, c$ (Å) | 118.3, 118.3, 249.9 | 92.8, 105.9, 164.2 |
| $\alpha, \beta, \gamma$ (°) | 90.0, 90.0, 120.0 | 90.0, 90.0, 90.0 |
| Wavelength (Å) | 0.8731 | 1.0723 |
| Resolution (Å)[a] | 79.2–2.6 (2.7–2.6) | 82.1–2.9 (3.0–2.9) |
| No. of copies per ASU | 2 | 1 |
| Total no. of reflections | 119,140 | 73,165 |
| No. of unique reflections | 60,031 | 36,612 |
| Completeness[a] | 98.83 (89.96) | 99.96 (100) |
| Multiplicity[a] | 2.0 (1.9) | 2.0 (2.0) |
| $I/\sigma I$[a] | 6.8 (1.4) | 9.1 (1.6) |
| $R_{merge}$ (%)[a] | 7.6 (51.08) | 2.8 (35.09) |
| $CC_{1/2}$ (%)[a] | 99.1 (50.2) | 100 (49.2) |
| *Refinement* | | |
| $R_{work}/R_{free}$ (%) | 19.8/24.7 | 20.9/25.1 |
| Average $B$-factor (A²) | | |
| Protein | 47.6 | 114.3 |
| R.m.s. deviations | | |
| Bond lengths (Å) | 0.003 | 0.002 |
| Bond angles (°) | 0.99 | 0.53 |
| Ramachandran plot | | |
| Favored (%) | 94.2 | 91.3 |
| Allowed (%) | 5.8 | 8.77 |
| Disallowed (%) | 0.0 | 0.0 |
| Rotamer outliers (%) | 0.4 | 0.97 |

$R_{free}$ value calculations are based on 5% randomly selected reflections. 5% of reflections were used for calculation of $R_{free}$
ASU: asymmetric unit, CC: correlation coefficient, R.m.s.d.: root mean square deviations
[a]Highest resolution shell in parentheses

antagonist of Wnt signaling[26,27] (Fig. 3d). Thus, similar to DKK, the VHHs act as a competitive inhibitor for Wnt3 ligands by sterically blocking access of these ligands to LRP6[P3].

**Inhibitory activity and specificity of anti-LRP5/6 VHHs**. Next, we characterized the activity and specificity of L-P2-B10, L-P2-D07 and L-P2-H07 VHHs in cell-based assays. Titration experiments revealed nanomolar range IC50 values for L-P2-B10, L-P2-D07 and L-P2-H07, for cellular responses to Wnt3a only (82, 94 and 106 nM, respectively) (Fig. 4a), as well as to the highly potent combination of Wnt3a and R-spondin1 (Rspo1) (40, 40 and 90 nM, respectively) (Fig. 4b). Thus, anti-LRP6 VHHs efficiently inhibit cellular responses to Wnt at nanomolar levels, even in the presence of the strong Wnt pathway agonist Rspo1. As expected, anti-LRP6 VHHs did not inhibit responses of HEK293T cells to CHIR-99021, a GSK3β inhibitor that compromises destruction complex activity to induce downstream β-catenin activation (Fig. 4c).

As LRP6 shares high homology with LRP5, we wondered whether the selected anti-LRP6 VHHs might also target LRP5-mediated signaling[37]. Sequence alignment of human LRP5[P3] and LRP6[P3] revealed strong conservation of the identified contact residues within the VHH-LRP6[P3E3P4E4] co-crystal structures (Supplementary Fig. 6), predicting sensitivity of Wnt3-mediated LRP5 signaling for targeted VHH treatment. To functionally validate these findings, we deleted *LRP6* from HEK293T cells using a CRISPR/Cas9-based strategy (Supplementary Fig. 7). Wnt3a-mediated pathway activation as well as cellular responses to LRP5 overexpression were fully blocked by anti-LRP6 VHHs in these *LRP6*⁻/⁻ cells (Fig. 4d). Thus, our VHHs target both LRP6 and LRP5. In further support of these findings, all three VHHs

strongly prevented Wnt3a-induced phosphorylation of endogenous LRP6 (S1490) (Fig. 4e, Supplementary Fig. 8).

Canonical Wnt ligands are divided in three main classes according to their interaction with different regions of the LRP5/6 extracellular domain[21,30,37]. While Wnt3 and Wnt3a selectively bind the P3 region in the third module (P3E3) (Fig. 1a), Wnt1 interacts with the first two modules of LRP6 (P1E1P2E2 region)[28]. We investigated the ability of the P3-binding VHHs to affect cellular responses to Wnt1. As expected, treatment with either control VHH or LRP5/6[P3]-binding VHHs did not affect Wnt1-induced reporter activity, confirming selective inhibition of Wnt3/3a-mediated Wnt/β-catenin pathway activation (Fig. 4f).

To investigate the ability of anti-LRP5/6[P3] VHHs to interfere with endogenous Wnt signaling, we employed small intestinal mouse organoids that critically depend on Wnt for viability. Organoids are embedded in matrigel and require supplementation with EGF, Noggin and Rspo (ENR medium) to drive tissue renewal in vitro[42]. In this setup, organoids fully recapitulate the crypt and villus structures and the associated proliferative and differentiated cell compartments present in the intestine[42]. When Wnt is provided exogenously, the self-established Wnt gradient of the tissue is lost, driving the organoid towards formation of cystic structures that are mainly characterized by proliferative cells[43]. We supplemented ENR-cultured small intestinal organoids with Wnt3a and either control or anti-LRP5/6[P3] VHHs. While control VHH-treated organoids displayed the expected Wnt-induced cystic morphology, treatment with anti-LRP5/6[P3] VHHs phenocopied the effect of Rspo withdrawal (EN medium), severely compromising cell viability (Fig. 4g). Collectively, these data show that anti-LRP5/6[P3] VHHs L-P2-B10, L-P2-D07 and L-P2-H07 efficiently block Wnt-mediated cellular activities in the intestinal epithelium. Moreover, our results confirm the vital role of endogenous Wnt3 for maintenance of intestinal organoids in vitro[44].

**Anti-LRP5/6 VHHs block *Rnf43/Znrf3*-mutant organoid growth**. Intestinal stem cells (ISC) locate to the bottom region of the crypt, where specialized niche cells create an optimal environment for stem cell maintenance[45]. ISCs are marked by the Wnt target gene *Lgr5*[46] and can self-organize and grow into three-dimensional organoids in vitro[42]. Inactivating mutations in *Rnf43* and *Znrf3* generate Wnt-hypersensitive stem cells that drive expansion of the intestinal stem cell zone and instigate adenoma formation in vivo[5]. Indeed, *Rnf43/Znrf3*-deleted (*R/Z* dKO) organoids grow independently of Rspo1 and thus display loss of niche factor requirement, a hallmark of tumorigenesis[5,47]. However, these *R/Z* dKO tumors remain dependent on the generation of Wnt3 by specialized Paneth cells in the tumor niche[15], revealing a vulnerability that might be targeted by anti-LRP5/6[P3] VHHs. Indeed, treatment with L-P2-B10 or L-P2-H07 induced massive cell death of tumorigenic *R/Z*-mutant organoids, comparable to treatment with the highly potent inhibitor of Wnt secretion, IWP-2[48] (Fig. 5a, b). Treatment with L-P2-D07 was less efficient and resulted only in a partial loss of cell viability (Fig. 5a, b).

To characterize the mechanism of VHH-mediated cell death, we investigated the possibility that blocking LRP5/6-mediated Wnt signaling depletes the self-renewing tumor cell pool by promoting their collective differentiation. Indeed, qRT-PCR analysis revealed that anti-LRP5/6[P3] VHH treatment induced loss of the stem cell markers *Lgr5, Olfm4* and *Axin2*, as well as the Paneth niche cell marker *Lys1*. At the same time, the differentiation markers *ChgA, Muc2* and *Alpi*[49] were strongly upregulated by VHH treatment (Fig. 5c, d). We confirmed these observations by confocal microscopy analysis, revealing loss of

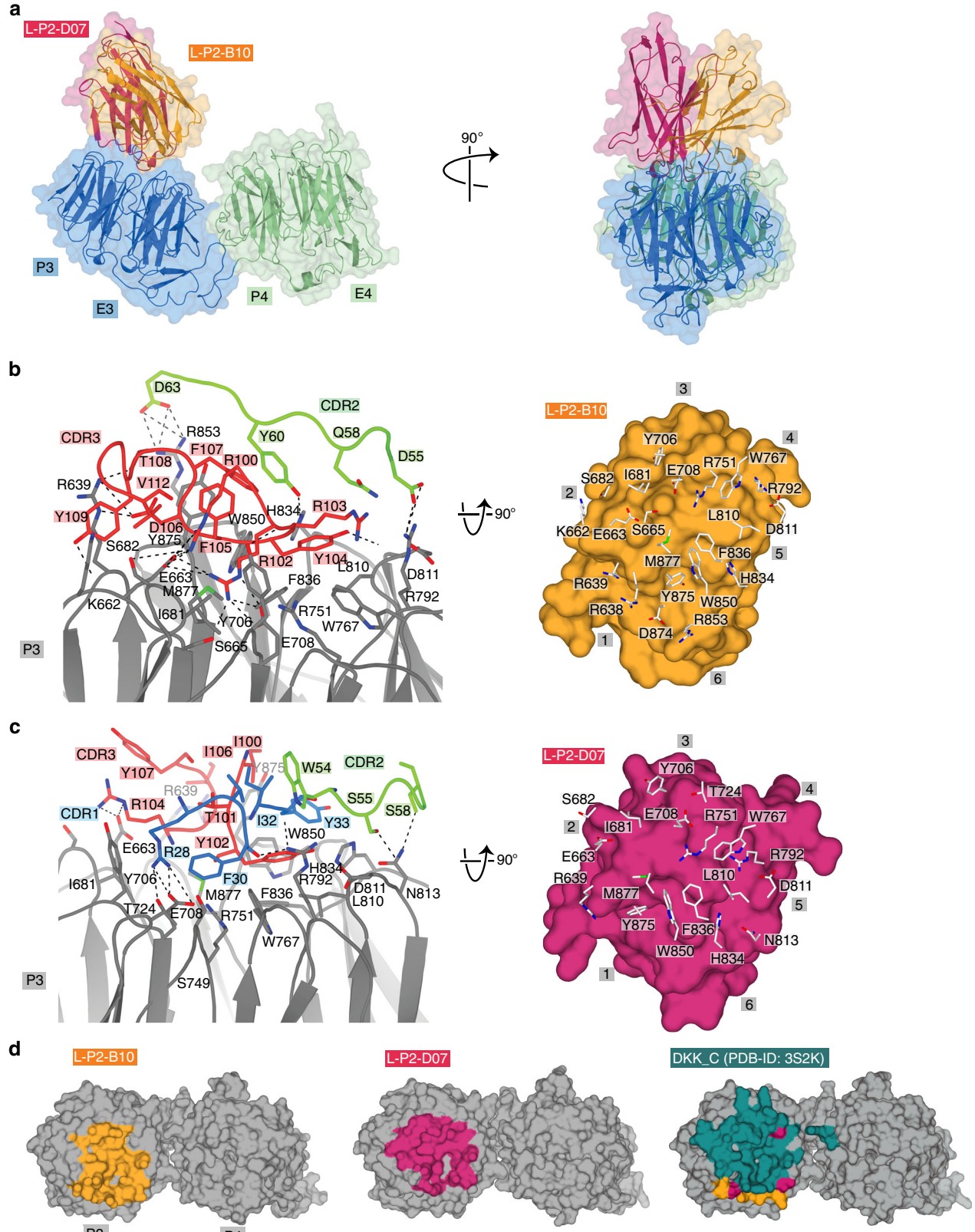

**Fig. 3** Structural analysis of L-P2-B10 and L-P2-D07 anti-LRP6 binding mode. **a** Alignment of the LRP6$^{P3E3P4E4}$ complexes with L-P2-B10 and LRP6–L-P2-D07 in two orthogonal views. Cartoon trace with transparent surface representation of LRP6$^{P3E3}$ (blue), LRP6$^{P4E4}$ (green), L-P2-B10 (orange) and L-P2-D07 (red). **b** Interface of LRP6–L-P2-B10 with on the left-hand side LRP6$^{P3}$ (gray) and CDR2 and CDR3 loops of L-P2-B10 (green and red, respectively). Interacting side chains are shown as sticks. On the right side, surface presentation of L-P2-B10 (orange) with interacting residues of LRP6$^{P3}$ shown in sticks; numbers represent the six blades of β-propeller 3. **c** Interface of LRP6–L-P2-D07 complex, presented similar to **b**, with in addition the CDR1 loop in blue (left) and the surface of L-P2-D07 in red (right). **d** Binding-surface areas of L-P2-B10 (orange), L-P2-D07 (red) and DKK-C[31] (magenta) on LRP6$^{P3E3P4E4}$ (gray)

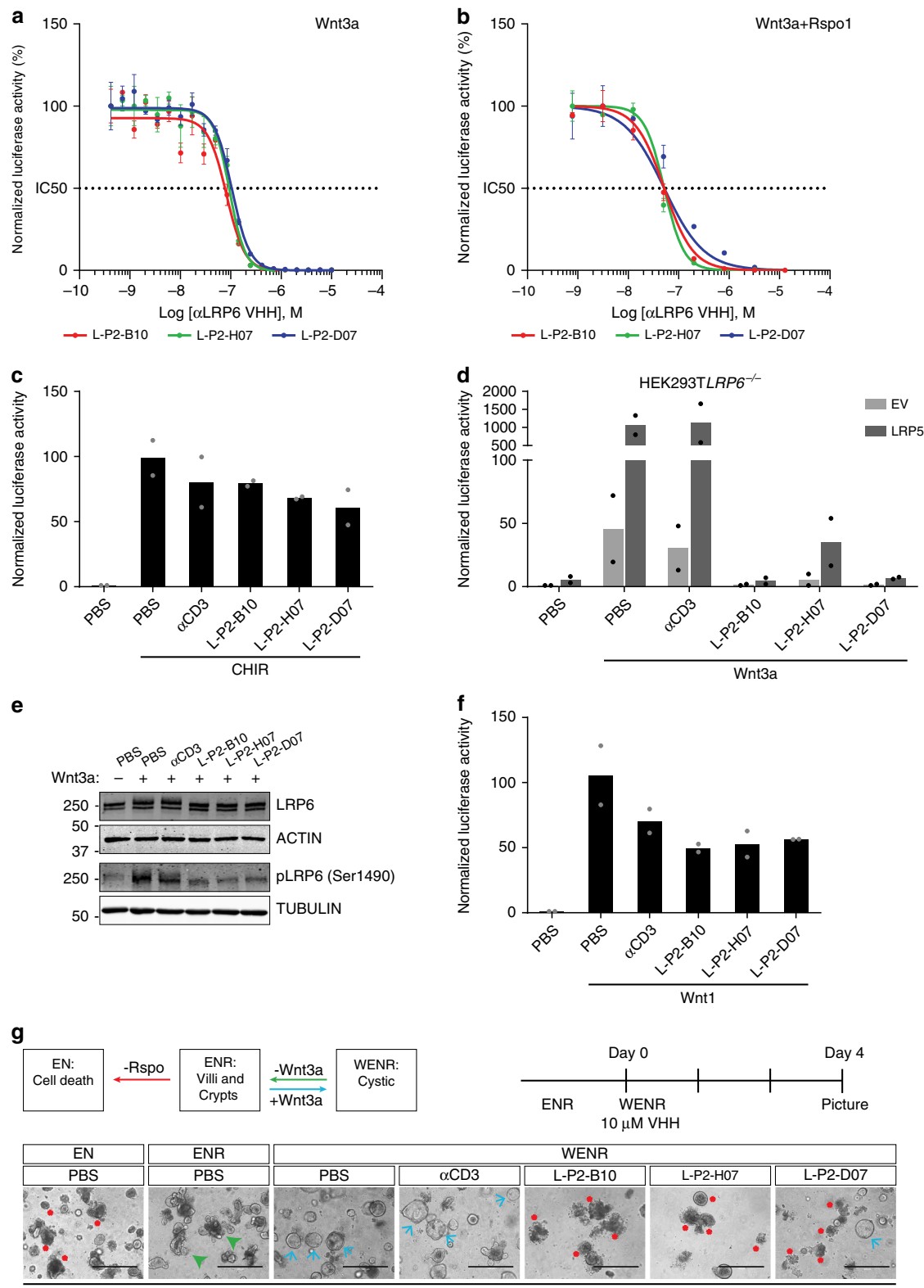

the proliferation marker Ki67 and the Paneth cell marker Lys, while the differentiation marker ChgA was increased upon VHH treatment (Fig. 5e). In conclusion, anti-LRP5/6^P3 VHHs effectively target Wnt-dependent tumorigenic organoids by removing an essential pathway for renewal of stem-like tumor cells and promoting their terminal differentiation.

## Discussion

Contrary to classical *APC* or *CTNNB1* mutations, *RNF43/ZNRF3* (*R/Z*) loss-of-function mutations drive a state of Wnt hypersensitivity, offering perspectives for therapeutic inhibition of receptor activity in subsets of patients suffering from such mutations[50]. Strikingly, murine R/Z^−/− tumors were eradicated by deletion of

**Fig. 4** Characterization of anti-LRP5/6 VHHs with highest potency. **a** IC50 calculations of inhibition of cellular responses to Wnt3a by titration of the indicated anti-LRP6[P3] VHHs in a luciferase reporter assay. Mean luciferase activities ± s.d. ($n = 3$) are plotted. **b** IC50 calculations by combination of Wnt3a and Rspo1. Mean luciferase activities ± s.d. ($n = 2$) are plotted. **c** Activation of Wnt signaling mediated by CHIR-99021 is not affected by anti-LRP6[P3] VHHs. Wnt luciferase reporter assay performed in HEK293T cells treated with 5 µM CHIR-99021 and 10 µM of the indicated anti-LRP6[P3] VHHs. Graph shows average (bars) and range (dots) of luciferase activity in duplicate cell cultures transfected in parallel. **d** Wnt luciferase reporter assay performed in LRP6[-/-] HEK293T cells. Wnt pathway activation achieved upon LRP5 overexpression is counteracted by anti-LRP5/6[P3] VHHs. Graph shows average (bars) and range (dots) of luciferase activity in duplicate cell cultures transfected in parallel. **e** Western blot showing inhibition of Wnt3a-induced LRP6 phosphorylation (pLRP6) by the different anti-LRP5/6[P3] VHHs. **f** anti-LRP5/6[P3] VHHs do not inhibit Wnt1-mediated cellular responses. Wnt luciferase reporter assay in HEK293T cells overexpressing Wnt1 and treated with 10 µM of the indicated anti-LRP5/6[P3] VHHs. Graph shows average (bars) and range (dots) of luciferase activity in duplicate cell cultures transfected in parallel. **g** Top panels: Schematic representation of the growth conditions, associated phenotypes and experimental set up. Bottom panels: anti-LRP5/6[P3] VHHs induce death of Wnt3a-treated wild type (WT) intestinal organoids. Organoids were cultured in WENR and treated with 10 µM of the indicated anti-LRP5/6[P3] VHHs for 4 d (Scale bar, 400 µm). Red asterisks indicate cell death; green arrows indicate organoids showing villi and crypts; blue arrows indicate cystic organoids

Wnt3, the primary Wnt produced by Paneth cells that co-populate the expanded tumor stem cell zone[15]. To target this R/Z[-/-] tumor vulnerability, we developed selective inhibitors for Wnt3/3a-mediated signaling. We used CIS display to select camelid single domain antibodies (VHHs) from a highly diverse, fully synthetic library that bind the P3E3P4E4 region of LRP6. This LRP6 region is involved in Wnt3-binding although the exact region and binding mode awaits identification[26,27,41]. From an initial panel of 33 LRP6[P3E3P4E4]-binding VHHs, we selected three VHHs that most efficiently inhibited cellular responses to Wnt3a at low nanomolar IC50 values. These VHHs efficiently blocked the growth of murine tumorigenic R/Z-mutant organoids.

To understand the structural basis by which anti-LRP6 VHHs exert their potent Wnt inhibitory activity, we generated crystal structures of L-P2-B10 and L-P2-D07 VHH in complex with the LRP6[P3E3P4E4] domain. The LRP6-binding epitope of both L-P2-B10 and L-P2-D07 covers a central region in P3 that overlaps with part of the ligand-binding surface of the natural antagonist DKK1_C[26,27]. Thus, as argued for DKK1, L-P2-B10 or L-P2-D07 operate by sterically preventing Wnt proteins from binding to LRP6[P3]. Furthermore, our results independently highlight this region in P3 as the core Wnt3-binding surface. Structural studies of various VHH-target complexes have revealed a large variability in length, conformation and usage of the three available CDR loops[51,52]. We show that L-P2-B10 employs a particularly long CDR3 loop that interacts with the VHH framework to create a stable flat paratope for recognition of a plane epitope at the center of the LRP6 P3 domain. CDR2 contributes in a minor way, while CDR1 does not participate in the interaction. By contrast, L-P2-D07 employs a much shorter CDR3 loop that, together with the CDR1 and −2 loops, interacts with the same relatively flat epitope.

Based on site-directed mutagenesis and antibody blocking experiments, separate binding sites for different Wnt isoforms in the extracellular domains of LRP6 were identified[21,22,26,28,30,41,53]. While Wnt3/3a proteins were classified as P3 binders, other Wnt isoforms such as Wnt1 and Wnt9b require a binding site in β-propeller 1 (P1). To antagonize LRP6 activity, DKK1 uses its N- and C-terminal domains to interact simultaneously with P1 and P3, respectively, thereby directly competing with binding of both classes of Wnts[26–28].

The relatively small DKK1_C fragment (88 aa) binds with high affinity to LRP5/6[P3E3P4E4] (Kd ~70 nM)[26], and is a promising candidate for Wnt inhibition[54]. However, DKK1_C was shown to interfere with the activities of Wnt1 and Wnt8[55,56] that operate via LRP6[P1E1P2E2] [21]. Indeed, biochemical data revealed that DKK1_C also interacts with LRP6[P1E1P2E2] (Kd ~70 nM)[27], thus compromising selectivity. Furthermore, multiple LRP6 antagonists, including DKK, WISE and SOST, as well as previously described LRP6-binding antibodies, make use of a short peptide

motif [Asn-X-Ile/Val] that mediates key interactions with LRP6[P1] [28]. This recognition motif is absent in the CDR sequence of our LRP6[P3]-binding VHHs, in line with their lack of interference with Wnt1-mediated signaling and apparent specificity for inhibition of Wnt3/3a-mediated cellular responses.

Two independent studies previously investigated the use of conventional monoclonal antibodies (mAbs) targeting LRP6. In both cases, the authors observed that the bivalent nature of mAbs significantly potentiates cellular responses to the non-blocked class of Wnts, likely due to induced receptor dimerization[21,30]. These unwanted side effects limit the possibility for clinical application of these mAbs, although a biparatopic anti-LRP6 antibody constructed by Ettenberg et al. might overcome these problems[30]. More recently, a bispecific antibody that combines two monospecific domain Abs was developed, targeting both the P1 and P3 binding sites on LRP6[31]. This broad-range Wnt inhibiting reagent however did not affect the growth of two Rspo-overexpressing patient-derived xenografts (PDX) models, presumably due to a lack of cross-reactivity with LRP5[31]. Here we show that, due to conservation of the VHH-binding epitope on LRP5[P3] and LRP6[P3], our VHHs overcome both these limitations, allowing for selective blockade of Wnt3/Rspo-mediated cellular responses while leaving non-blocked Wnts unaffected.

Within the healthy intestinal epithelial stem cell niche, locally produced Wnt/Rspo signals ensure highest levels of Wnt activity required to preserve stem-cell properties and the generation and maintenance of specialized Paneth cells[44]. Loss of R/Z-mediated negative feedback due to inactivating mutations drives an Rspo-independent Wnt-hypersensitive state that mediates expansion of both stem and Paneth cell populations and the formation of adenomas in vivo[5,15]. This tumorigenic step is recapitulated in vitro by R/Z-mutant tumor organoids that grow independently of Rspo supplementation[5,15]. Treatment with anti-LRP5/6[P3] VHHs strongly blocked the growth of murine Wnt-hypersensitive tumorigenic R/Z-mutant organoids. Importantly, impaired viability upon VHH treatment was not due to direct toxic effects but was accomplished through exhaustion of the stem cell pool via induction of collective terminal differentiation. At the same time, Paneth cells, the main constituents of the intestinal stem cell niche, were depleted[57]. These findings show that VHH treatment not only blocks cellular responses to Wnt3 but also removes the main source of Wnt production. Thus, anti-LRP5/6[P3] VHH treatment phenocopies the genetic ablation of Paneth cells or Wnt3, which was shown to prevent R/Z dKO tumorigenesis in mouse models[5].

The induction of collective tumor cell differentiation by Wnt signaling suppression has emerged as a promising strategy for the eradication of CRC[13,58,59]. A major challenge for targeting Wnt signaling in cancer, however, is the identification of efficacious agents that reduce Wnt signaling in tumor cells without

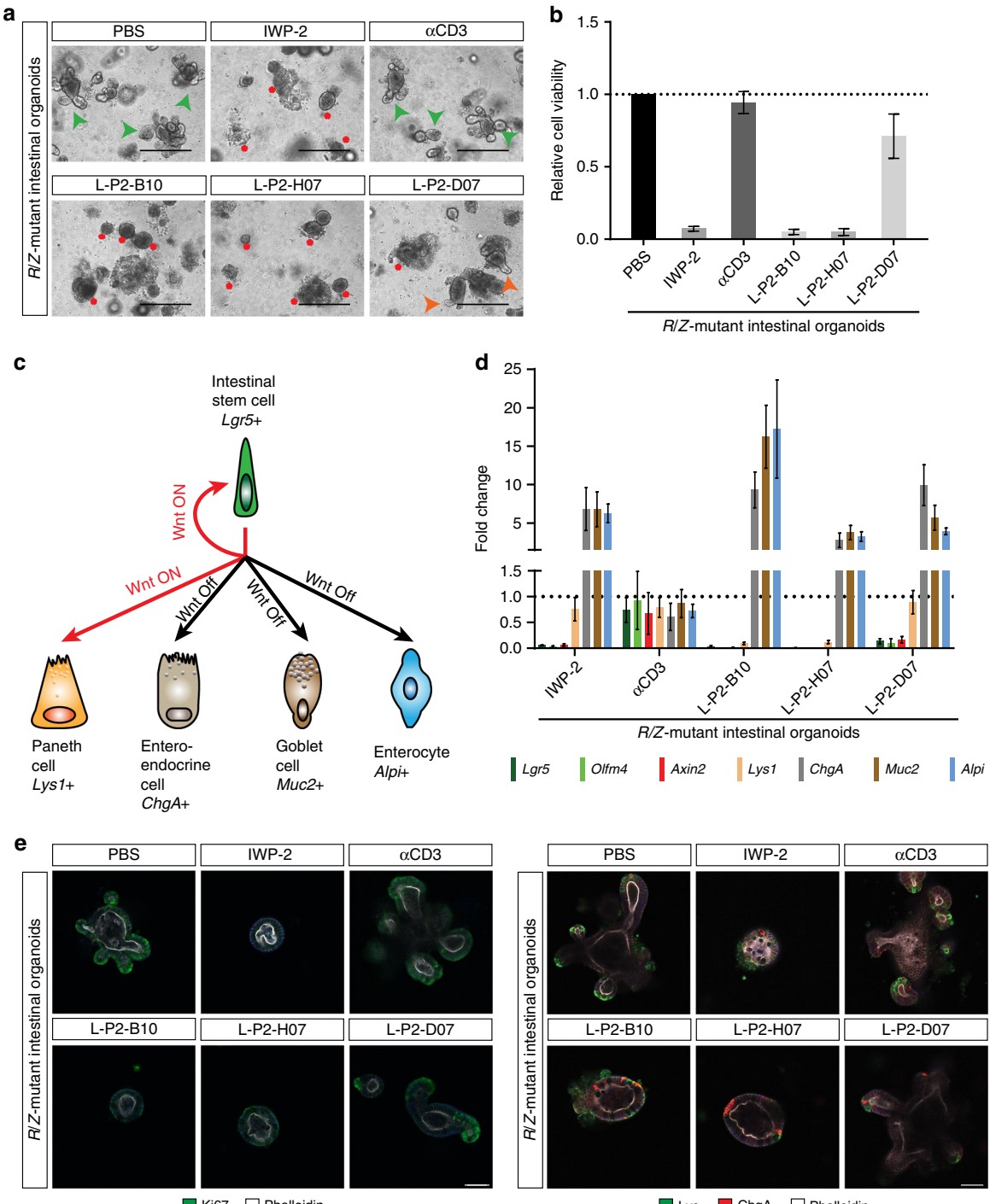

**Fig. 5** Anti-LRP5/6 VHHs drive collective differentiation of R/Z-mutant tumor organoids. **a** Anti-LRP5/6[P3] VHHs block tumorigenic R/Z mutant organoid growth. Organoids were cultured in EN and treated with 10 µM of the indicated anti-LRP5/6[P3] VHH for 4 d (Scale bar, 400 µm). Red asterisks indicate cell death; green arrows indicate organoids showing villi and crypts; orange arrows indicate organoids showing a mixed phenotype of cell death and villi crypts structures. **b** Anti-LRP5/6[P3] VHHs strongly diminish cell viability of tumorigenic R/Z mutant organoids. Organoids were cultured in EN and treated with 10 µM of the indicated anti-LRP5/6[P3] VHHs for 4 d. Graph represents relative cell viability normalized to PBS treatment. Mean ± s.d. (n = 3) are plotted. **c** Schematic representation of the cell types present in the intestine and the role of Wnt signaling in maintaining intestinal physiology. **d** qRT-PCR showing anti-LRP5/6[P3] VHH treatment (3 d) strongly inhibits the expression of Wnt target genes, stem cell-associated genes and Paneth cell markers while inducing a transcriptional program for differentiation in tumorigenic R/Z mutant organoids. Graph represents fold change in gene expression as compared to PBS treatment (= 1). Mean ± s.d. (n = 3) are plotted. **e** Anti-LRP5/6[P3] VHH treatment (3 d) attenuates proliferation as shown by a decrease in Ki67 staining (left). Anti-LRP5/6[P3] VHH treatment decreases the number of Paneth cells (Lys) while increasing the number of differentiated enteroendocrine cells (ChgA) (right). Phalloidin staining was used to mark the organoids (Scale bar, 10 µm)

damaging normal stem cell function and tissue repair[10,16]. Inhibitors that directly target production of all Wnts are under development for cancer therapy but may not be well tolerated due to severe side effects on tissue regeneration and repair[10,16]. More selective approaches may offer a viable alternative, as shown recently by blockade of RSPO3, which inhibited tumor growth of PTPRK-RSPO3 fusion-expressing CRC by promoting differentiation and loss of stem-cell function[13]. Our results demonstrate that VHH-mediated interference with selective Wnts involved in canonical Wnt signaling branches holds promise for targeting Wnt-hypersensitive tumors, while potentially limiting side effects. Due to their small size, VHHs can rapidly and deeply penetrate solid tumors, although their activity is limited by their short half-life in vivo[34,60]. Therefore, strategies will be required to extend the half-life of these anti-LRP5/6 VHHs before moving towards in vivo applications[61]. The broad species cross reactivity of these VHHs as anticipated based on in silico modeling of the contact surfaces (rodent, non-human primate) should facilitate both therapeutic proof-of-concept experiments as well as toxicology studies.

## Methods

**Cell culture and reagents**. Human embryonic kidney (HEK)293T cells (ATCC CRL-3216) were cultured in RPMI medium (Thermo Scientific) supplemented with 10% Fetal Calf Serum (Bodinco B.V.), 100 units/mL penicillin and 100 µg/mL streptomycin (Thermo Scientific). Mouse L-cells were cultured in DMEM containing 1 g/L glucose (Thermo Scientific) with the same supplements. L-cells stably expressing Wnt3a were used to generate Wnt3a-conditioned medium (Wnt3a-CM). R-spondin1-CM and Noggin-CM were produced using HEK293T cells stably transfected with human Rspo1-V5 or after transient transfection with mouse Noggin-Fc expression vector, respectively. All cells were grown at 37 °C at 5% CO$_2$. CHIR-99021 (Tocris bioscience) was used at 5 µM for 5 h.

**Small intestine mouse organoids**. Small intestine mouse organoids were established and maintained as described[42] from isolated crypts collected from the entire length of the small intestine. The basic culture medium (advanced DMEM/F12, with penicillin/streptomycin, 10 mM HEPES, 1× Glutamax, 1× B27 [all from Life Technologies] and 1 mM N-acetylcysteine (Sigma Aldrich) was supplemented with 50 ng/mL murine recombinant EGF (Peprotech), Noggin-CM (1% v/v) and R-spondin1-CM (2.5% final volume, if not indicated otherwise) to obtain ENR medium. Wnt3a-CM was used at 50% (v/v) to obtain WENR. The Rnf43/Znrf3 double knock-out (R/Z dKO) organoid line was a gift of BK Koo, Wellcome Trust - Medical Research Council Stem Cell Institute, University of Cambridge, Cambridge, UK[62] and grown in EN medium (ENR without Rspo1). Organoids were cultured in matrigel droplets (Corning).

**DNA constructs and CRISPR targeting**. Myc-LRP6 was a gift of C. Niehrs, Institute of Molecular Biology, Mainz, Germany. LRP5-Myc-His was a gift of G. Bu, Mayo Clinic, Jacksonville, United States. mWnt1 was a gift of H. Clevers, Hubrecht Institute, The Netherlands. LRP6 CRISPR-Cas9 targeting construct was generated by ligating the annealing product of the primers listed in Supplementary Table 1 to pX330 (obtained from Addgene #48139) digested with BbsI (Thermo Scientific). Single clones were established and genotyped. For genotyping, genomic DNA was isolated using QIAamp DNA micro kit (Qiagen). Primers for the PCR amplification using GoTaq Flexi DNA polymerase (Promega) are listed in Supplementary Table 1. Products were cloned into pGEM-T Easy vector system I (Promega) and subsequently sequenced using a T7 sequencing primer.

**Immunoblotting**. Western blotting was performed using standard procedures with Immobilon-FL (Milipore) PVDF membranes. For blocking the Odyssey blocking buffer (LI-COR) was used and immunoblot analysis was performed using the LI-COR Odyssey Infrared Imaging System.

**Antibodies**. For Immunoblotting: mouse anti-α-Tubulin (Sigma Aldrich, T5168, 1:10,000); mouse anti-actin (MP Biomedicals (CLONE C4), 08691002, 1:10,000); rabbit anti-LRP6 (Cell Signaling (C5C7), #2560, 1:1000); rabbit anti-Phospho-LRP6 (Ser1490) (Cell Signaling, #2568, 1:500). All goat secondary antibodies were conjugated with either Alexa Fluorphores (Life Technologies, A-21109, 1:8000) or IRDyes (LI-COR, P/N 926–32210, 1:8000). For immunofluorescence: rabbit anti-Ki67 (Abcam, 15580, 1:100), rabbit anti-Lysozyme (Dako, A0099, 1:500), mouse anti-ChgA (Santa Cruz (C-12), sc-393941, 1:200). As secondary antibodies goat anti-rabbit IgG-488 (Life Technologies, A-21441, 1:300) and goat anti-mouse-IgG-568 (Life Technologies, A-11004, 1:300) were used. Hoechst 33342 (Sigma, B-2261) was used at 1:500 and Phalloidin (Ph647) (Cell Signaling, #8940) at 1:50.

**Luciferase reporter assay**. Luciferase TOPFlash and FOPFlash reporter assays were performed as described[63]. Briefly, HEK293T cells were seeded in 96 well plates (22,000 cells/well) and transfected with 5 ng TopFlash[64] reporter and either empty vector or 5 ng of myc-LRP6 or 7.5 ng of LRP5-Myc-His. For the autocrine Wnt1 production, 0.3 ng of mWnt1 was co-transfected. Transfection efficiency was controlled and normalized by including 1 ng of TK-Renilla reporter plasmid in all transfections. All transfections were performed with FuGENE 6 transfection reagent (Promega) according to the manufacturer's protocol. Six hours post transfection, cells were incubated with 10 µM of VHH and either control conditioned medium or Wnt3a-CM overnight. Luciferase activity was measured using the Dual Luciferase Reporter Kit (Promega) according to the manufacturer's protocol in a Centro XS 960 microplate luminometer (Berthold). Values were normalized to untreated controls.

**Microscopy**. For bright field microscopy organoids were grown in 10 µL matrigel droplets and images were captured with an EVOS (Thermo Scientific). For immunofluorescence microscopy, R/Z dKO organoids were grown in 10 µL matrigel in slide angiogenesis chambers (Ibidi) in 40 µL EN and treated with 10 µM of IWP-2 (R&D systems) or 10 µM of each VHH for 3 d. After treatment organoids were fixed in paraformaldehyde (4% diluted in 0.1 M Na-Pi) for 1 h after which paraformaldehyde was removed and 20 mM NH$_4$Cl (in PBS) was added for 10 min. Organoids were permeabilized in PBD 0.2 T buffer (1% BSA, 1% DMSO, 0.2% TX-100 in PBS) for 2 h at room temperature (RT). Organoids were incubated overnight with primary antibody at 4 °C and 3 h with secondary antibody including HOECHST at RT in PBD 0.2 T. Organoids were mounted in Ibidi mounting medium and images were acquired with a Zeiss LSM700 confocal microscope. Images were processed with ImageJ.

**Viability assay**. Organoids were trypsinized in the presence of Y-27632 (Tocris) and 1000 single cells per well were seeded in a 96-well plate in Basement Membrane Extract (BME) (Cultrex). Before addition of 100 µL of EN media, the BME was polymerized for 20 min at 37 °C. Organoids were cultured for the indicated times in the absence or presence of the indicated VHHs. The Cell titer Glo Luminiscent Cell viability assay (Promega) was used to assess viability of the organoids in a Centro XS 960 microplate luminometer.

**Reverse transcription and quantitative real-time PCR**. RNA was extracted using the Qiagen RNeasy kit (Qiagen) and used as a template for cDNA production using the iScript™ cDNA Synthesis Kit (Biorad) according to the manufacturer's protocol. The synthesized cDNA was subsequently used in a qRT–PCR using IQ SYBR green mix (Biorad) according to the manufacturer's protocol. The ΔΔCt method[65] was employed to calculate gene expression. The primer pairs used for the qRT-PCR are listed in Supplementary Table 2

**Expression and purification of LRP6 ectodomain**. Human LRP6 containing domains P3E3P4E4 (Uniprot residues 639–1244) was cloned in the expression vector pUPE107.03 (U-Protein Express BV, The Netherlands) with an N-terminal cystatin secretion signal and a C-terminal His$_6$-tag. LRP6$^{P3E3P4E4}$ was transiently co-expressed in HEK293 deficient in N-acetylglucoaminyltransferase I (GnTI-) cells with human MESD. Six days after transfection the conditioned medium was harvested by centrifugation for 20 min at $1000 \times g$. LRP6 was purified by the addition of Ni-Sepharose excel resin (GE Healthcare) and incubated overnight at 4 °C. A packed column was washed using immobilized metal ion affinity chromatography (IMAC) A buffer (25 mM HEPES pH 8.0, 500 mM NaCl and 10 mM Imidazole). Proteins were eluted in IMAC B (25 mM HEPES pH 8.0, 500 mM NaCl and 250 mM Imidazole), followed by a purification step with size-exclusion chromatography (SEC), using a Superdex 200 HiLoad 16 60 column (GE Healthcare) in 15 mM HEPES, pH 7.4 and 150 mM NaCl and concentrated to 13 mg/mL. To produce biotinylated human LRP6$^{P3E3P4E4}$, it was cloned in an expression vector (U-Protein Express BV, The Netherlands) with a C-terminal BAP-tag followed by a His$_6$-tag. LRP6$^{P3E3P4E4}$ was transiently co-expressed in HEK293 GnTI-cells with human MESD and Bir-A. Five days after transfection 0.2 µM Biotin (Sigma Aldrich) was added to the medium and incubated overnight to obtain biotinylated LRP6$^{P3E3P4E4}$. Purification was performed as described for the non-biotinylated LRP6$^{P3E3P4E4}$.

**CIS display selection**. CIS display selections were carried out as described[36,66]. In total four different VHH sublibraries were displayed to identify LRP6-specific binders that explored different CDR3 loop lengths ranging from 7 to 22 residues. Briefly, 6.5 µg of linear template DNA encoding for the VHH libraries in CIS display format was added to 200 µL in vitro transcription/translation (ITT) reaction mixture (Isogenica) and incubated for 1 h at 30 °C. The ITT reaction was stopped by addition of 400 µL 2% BSA in PBS and incubation on ice for 10 min. A deselection step was performed by adding 100 µL of M-280 Streptavidin Dynabeads (Thermo Scientific) (pre-blocked in 1% BSA in PBS) and incubation for 1 h at RT with mixing. After removal of the magnetic beads the supernatant was transferred to 50 µL of M-280 Streptavidin Dynabeads pre-coated with 21.5 µg of biotinylated LRP6$^{P3E3P4E4}$ and 1% BSA in PBS. The resulting samples were incubated for 30 min at RT with mild mixing to allow binding of the CIS displayed VHH to

LRP6, after which the beads were washed 5 times with PBS-T (PBS, 0.1% Tween-20) for 2 min, followed by a final wash in PBS. Bound DNA was eluted from the beads by incubation in 1x ThermoPol buffer (New England Biolabs) for 10 min at 80 °C. The eluted material was added to a recovery PCR reaction[67] and finally purified using a Wizard® SV Gel and PCR Clean-up System (Promega).

For subsequent rounds of expression and selection, the purified DNA amplification products from the preceding round was added to a fresh ITT mixture and the selection process was repeated 5 times to enrich the DNA pool for binding clones. To increase the stringency of the selection, the amount of antigen was reduced each round (from 21.2 to 2.7 µg) while the number of washing steps of target/library complex-coated beads was simultaneously increased from 5 to 11. Progression of the selection was monitored by qPCR using a StepOne Plus system (Thermo Scientific) and KAPA SYBR Fast ABI Prism (Sigma Aldrich). Template DNA was measured by qPCR both before the ITT (input DNA) and after target pull-down (output DNA) following ITT and panning on target-coated beads. A primer pair specifically amplifying a common region of the RepA gene (Supplementary Table 3) was used to determine the efficiency of template DNA recovery at various rounds of selection.

**ELISA screening.** After five rounds of selection, the recovered DNA was PCR-amplified with Phusion Green HF (Thermo Scientific) using primers TAC6 and NOTIRECREV (Supplementary Table 3) to amplify the VHH segment and introduce a C-terminal V5-tag and NotI site downstream of the VHH coding sequence. Amplified DNA was purified, digested with NcoI and NotI (New England Biolabs) and ligated into a pET vector containing additional polyHis- and FLAG-tags C-terminal of the insert. Ligations were transformed into E. coli BL21 Gold Competent cells (Agilent, Stockport, UK) and plated on 2YT media plates containing 50 µg/mL kanamycin (Sigma Aldrich). Individual colonies were picked and grown for soluble expression of VHH.

Briefly, induction of 1 mL cultures was started using 1 mM IPTG (Sigma Aldrich) in cultures grown to OD$_{600}$ ~1, followed by further overnight incubation at 30 °C. Cells were lysed by snap-freeze and thaw followed by treatment with Bugbuster Master Mix (Merck) according to the manufacturer's instructions. The unpurified, soluble supernatant fraction was directly used in ELISA screening. ELISA screening was carried out on NUNC Maxisorp plates coated with 250 ng per well of streptavidin, followed either by overnight incubation at 4 °C with 100 ng/ well of biotinylated LRP6$^{P3E3P4E4}$ (target wells) or PBS buffer (negative control wells). Horseradish peroxidase-conjugated anti-V5 secondary antibody (Abcam), diluted 1: 3000 in 2% BSA in PBS was used to detect VHH/anti-V5 binding. Binding was detected using SureBlue TMB peroxidase substrate (Insight Biotechnology), with plates being read at 450 nm. Clones showing an OD > 0.4 on LRP6 and specificity over the negative control wells (LRP6/Streptavidin ration > 5) were selected for a confirmatory ELISA, performed according to the same protocol. Clones with confirmed binding were sequenced. Sequences were aligned and correlated to values from the confirmatory ELISA (LRP6 signal/streptavidin signal) to select the best unique sequence hit clones.

**Expression and purification of VHH domains.** Selected VHHs were expressed in E. coli BL21 cells (BL21-Gold Competent Cells, Agilent) grown in Terrific Broth (TB) (Sigma Aldrich) containing 50 µg/mL kanamycin. Induction and lysis was performed as described above. C-terminal V5-FLAG$_3$-His$_6$-tagged VHH material was purified from cell lysates by IMAC, either using His MultiTrap HP plates or HisTrap 1 and 5 mL columns (GE Healthcare Life) in case of purifications in 96 well format from 1 mL cultures or from large scale cultures, respectively. Protein-containing fractions were pooled and imidazole was removed by either buffer exchange (PD Multitrap G-25) or extensive dialysis (snakeskin dialysis tubes MWC 10 kDa, Invitrogen). The final purification step (using PBS) was performed with size exclusion chromatography (SEC) using a Superdex 75 HiLoad 16 60 column (GE Healthcare Life Sciences).

For initial crystallization screens, binding experiments and in vitro assays, VHHs with a C-terminal V5-FLAG$_3$-His$_6$-tag were used. For optimized crystallization, VHHs L-P2-B10 and L-P2-D07 were cloned into the pETX-24 vector containing a C-terminal His$_6$-tag. The VHHs were expressed in E. coli BL21 (DE3)pLysS cells (Agilent) by lactose auto-induction[68]. VHHs were extracted by sonication in PBS. Cellular debris was removed by centrifugation for 30 min at 8000 × g and filtered with a glass fiber prefilter (Ministart, Sartorius). Imidazole was added to a final concentration of 15 mM. VHHs were purified using Ni-Sepharose 6 Fast Flow beads (GE Healthcare). Fractions containing protein after elution with 300 mM imidazole were concentrated and further purified by SEC using a Superdex 75 Hiload 16 60 column in 15 mM HEPES, pH 7.4 and 150 mM NaCl and concentrated to 10 mg/mL.

**Isothermal titration calorimetry.** Isothermal titration calorimetry (ITC) was performed using a MicroCal Auto ITC200 (Malvern Instruments Ltd). A 10 µM solution of LRP6$^{P3E3P4E4}$ in DPBS was placed into the 200 µL sample cell at 25 °C. Titration was performed with 2 µL injections (first injection 0.5 µL) of VHHs in DPBS at a concentration of 100 µM every 120 s (26 injections total). Data were fitted using the MicroCal PEAQ-ITC Analysis Software (Malvern Instruments Ltd) according to standard procedures. Fitted data yielded the stoichiometry (n), the

dissociation constant (kD), enthalpy (H) and entropy (S). Each ligand test was performed in triplicate and values for n, kD, H and S represent mean ± s.d. of three independent experiments.

**QCM data collection and analysis.** Quartz crystal microbalance (QCM) technology was used to analyze the affinities of the VHHs towards LRP6. The experiments were performed using an Attana A200 C-Fast system (Attana AB). LRP6$^{P3E3P4E4}$ was immobilized on a low-non-specific binding chip (LNB) via amine coupling according to the manufacturer's protocol. Two sensor chips were docked in the instrument and the temperature was set to 22 °C. HBS-T was passed over the surfaces at a flow rate of 100 µL/min until the baseline was stabilized (frequency change ≤ 0.2 Hz over 600 s). After reduction of the flow rate to 10 µL/ min, a freshly prepared activation solution (0.2 M 1-ethyl-3-(3-dimethylamino-propyl)-carbodiimide, 0.05 M sulfo-N-hydroxysuccinimide) was injected for 300 s on both surfaces. Then 50 µg/mL of LRP6$^{P3E3P4E4}$ in sodium acetate buffer, pH 4.5, was injected on one surface for 300 s. Finally, a de-activation solution consisting of 1 M ethanolamine, pH 8.5, was injected for 300 s. Immobilization of LRP6$^{P3E3P4E4}$ caused a frequency shift of 200 Hz in channel A.

The flow was set to 25 µL/min and the experiment was initiated after baseline stabilization. VHHs diluted in HBS-T at four concentrations (0.25–0.13–0.06–0.03 µg/mL) were injected sequentially for 84 s in parallel on both surfaces. In a single-cycle-kinetics approach starting at the lowest concentration, each injection was followed by 30 s dissociation and followed by injection of the next higher concentration. Finally dissociation was observed by injection of HBS-T for 300 s. After each cycle the surface was regenerated with an 84 s pulse of 10 mM glycine, pH 3.5. All VHHs were injected in duplicate in two independent experiments. Data were collected using Attester software (Attana AB) and analyzed with Evaluation software (Attana AB) and Clamp XP (University of Utah). The signal obtained from the reference surface was subtracted from the sensograms obtained for the LRP6$^{P3E3P4E4}$− immobilized surface. To calculate the kinetic parameters ($k_a$ and $k_d$) and affinity constant ($K_D$), the experimental data were fitted using a homogeneous ligand model (1:1 fitting model).

**Crystallization of LRP6–VHH complexes.** Purified LRP6$^{P3E3P4E4}$ was mixed with purified L-P2-B10 (C-terminal His$_6$-tag) or VHH L-P2-D07 (C-terminal V5-FLAG$_3$-His$_6$-tag), respectively in a 1:1.05 receptor:VHH ratio and incubated for 2–4 h at 4 °C. The final concentration of the mixed receptor:VHH complexes was 131 and 145 µM, respectively. Crystals of the LRP6$^{P3E3P4E4}$:L-P2-B10 complex grew at 18 °C as long needles by sitting drop vapor diffusion technique equilibrating the protein mixture with a reservoir solution containing 0.1 M sodium citrate, 0.2 M sodium acetate trihydrate, pH 5.5 and 10% PEG w/v 4000 in a 1:1 protein:mother liquor ratio. Crystals were cryo-protected by briefly taking them up in mother liquor supplemented with 25% (v/v) ethylene glycol before flash-freezing in liquid nitrogen. Crystals of the LRP6$^{P3E3P4E4}$:L-P2-D07 complex grew by sitting drop vapor diffusion in a 1:1 protein:mother liquor ratio with a reservoir solution containing 0.1 M MES, pH 5.0 and 10% (w/v) PEG 6000. Crystals were transferred into cryo-protectant solution of mother liquor supplemented with 20% (v/v) glycerol and flash frozen in liquid nitrogen.

**X-ray crystallography data collection and complex refinement.** LRP6$^{P3E3P4E4}$ -L-P2-B10 crystals were collected by helical collection and diffracted to 2.6 Å resolution at beamline ID 23–2 at European Synchrotron Radiation facility (ESRF Grenoble, France). X-ray diffraction data for complex LRP6$^{P3E3P4E4}$–L-P2-D07 was collected at beamline ID29 at ESRF and diffracted to 2.9 Å resolution, see Table 1. X-ray diffraction data for both complexes were integrated with DIALS[69] and scaled using Aimless[70] in the CCP4 suite[71]. Data processing statistics are shown in Table 1. Phasing of both datasets was performed by molecular replacement using program PHASER[72], by previously solved structures of LRP6 (PDB ID: 4A0P[41]) and VHH framework without CDRs (PDB ID: 4KRN[73]) as search models. In all VHH domains, sufficient density was available to model the truncated CDR loops. All structures were iterative refined with Phenix-Refine[74] and alternated with manual model building and model improvement using coot[75]. Statistics of diffraction data processing and refinement for both complexes are summarized in Table 1. Molprobity[76] was used for structure validation. Figures were generated with PyMOL (The PyMOL Molecular Graphics System, Version 2.0 Schrödiger, LLC).

**Reporting Summary.** Further information on experimental design is available in the Nature Research Reporting Summary linked to this Article.

## Data availability

The coordinates and structure factors have been deposited in the Protein Data Bank under accession codes 6H15 and 6H16. Other data are available from the corresponding author upon reasonable request. The source data underlying Figs. 1b, c, 2a–c, 4a–f, 5b–d and Supplementary Figure 2 is provided as a Source Data file.

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

## Acknowledgements

We thank members of the laboratory of M.M.M. for experimental support, helpful discussions and suggestions. We thank the European Synchrotron Radiation Facility (ESRF) for the provision of synchrotron radiation facilities; and beamline scientists of the ESRF for assistance. We thank U-protein Express for the provision of protein expression facilities. We thank Guangyun Yu and Ronald Pieters (Utrecht University, Department of Pharmaceutical sciences - Molecular Pharmacy, The Netherlands), for providing access to the MicroCal Auto ITC200. This work is part of the Oncode Institute, which is partly financed by the Dutch Cancer Society. This work was supported by European Research Council Starting Grant 242958, the Netherlands Organization for Scientific Research NWO VICI Grant 91815604, ECHO Grant 711.013.012 (to M.M.M.), European Union Grant FP7 Marie Curie ITN 608180 "WntsApp" (to M.M.M.), Fondazione Michelangelo for the Advancement of the Study and Treatment of Cancer (to N.F.), BioStruct-X Grant 283570 and NWO Spinoza Grant 01.80.104.00 (to P.G.).

## Author contributions

N.F., R.C.v.S., C.U., P.G. and M.M.M. conceived and designed the experiments. N.F., R.C.v.S., M.G., D.P., I.J., T.K., S.S. and T.Z.B. performed the experiments. N.F., R.C.v.S., M.G., D.P., I.J., S.S., T.Z.B., P.G. and M.M.M. analyzed the data. G.H., C.U. and T.A. contributed reagents and analytic tools. N.F., R.C.v.S., M.G., I.J., D.P., P.G. and M.M.M. wrote the manuscript, which was reviewed by all authors.

## Additional information

**Competing interests:** A patent application covering this work has been filed by UMCU and Isogenica, naming M.G., C.U., N.F. and M.M.M. as inventors. The remaining authors declare no competing interests.

