## [Peer Review File · Nature Communications]

Reviewers' comments:

Reviewer #1 (Remarks to the Author):

Fenderico et al. report the discovery of novel single-domain antibody fragments (VHHs), which specifically inhibit Wnt3a through binding to the Dkk_C- (and likely also the Wnt3a-) binding surface of the LRP6 P3 domain. Biochemical, structural and functional data are all consistent with inhibition through specific competitive binding with Wnt3a by VHHs. Given the role of Wnt3a in cancer development, achieving Wnt3a-specific inhibitors that can block the growth of Rnf43/Znrf3-mutant intestinal organoids is quite significant. Data presented in this manuscript, including the crystal structure, are of high quality. Overall, I feel that this manuscript, upon appropriate revision, is suitable for publication in Nature Communications.

Specific issues:

1. What is the advantage of these VHHs over Dkk_C in potential biomedical use? Authors discussed the undesirable lack of Wnt(s)-inhibitory specificity of previously reported LRP6 antibodies. It would be nice to discuss potential issues with human Dkk_C domains, which can bind to the LRP6 P3 domain with nanomolar affinity and should not have any immune problem.
2. Why do all VHHs have two binding affinities with LRP6 P3E3P4E4? What is the implication for Wnt inhibition here? Can authors cross-validate this two-Kd-value phenomenon using other biophysical techniques (e.g. ITC) for the three best-studied VHHs?
3. Some areas of this manuscript are repetitive.
4. (minor) Can authors fix the structure of the Ramachandra Plot outlier residue (Table 1)?
5. (minor) What does the rotation arrow between Figs 3a and 3b mean? Do both Figs 3b and 3d have the same orientation, which is orthogonal with that of 3a?

Reviewer #2 (Remarks to the Author):

I have reviewed the paper "Anti-LRP5/6 VHHs promote differentiation of Wnt-hypersensitive intestinal stem cells". The major claims of the paper are the identification of single domain antibody fragments that selective block Wnt3/3a ligand engagement and signaling through LRP5/6. The uniqueness of the molecules identified are the ability to block Wnt3/3a signaling without affecting Wnt1-class ligand engagement and signaling. In addition, the molecules are effective against both LRP5 and LRP6, thereby addressing receptor redundancy. These properties are advantageous for potential treatment of colorectal cancers (CRCs) activated by mutations in RNF43, ZNRF3, RSPO2 and RSPO3.

The authors convincingly demonstrate activity of the molecules. Using a reporter assay, they identify functionally active molecules, performed structural studies and validate activity in the LRG5 mouse organoid model. A minor drawback is lack of in vivo data, although the authors admit that further improvement is needed for in vivo studies.

Overall, this is a very well-done and enjoyable paper to read. Great job by the authors - I didn't find a single typo. The figures are clear and well-presented. The scientific results provide important insights into LRP5/6 structure/function relationships that will assist in the development of therapeutics targeting CRC. While in vivo studies would have value to the manuscript, I feel that it is still suitable for Nature Communications and recommend publication as is.

Reviewer #3 (Remarks to the Author):

In the manuscript entitled "Anti-LRP5/6 VHHs promote differentiation of Wnt-hypersensitive intestinal stem cells", Fenderico and colleagues describe a novel single-domain antibody that affects the Wnt3/3a signaling at nanomolar scale. They also showed the efficacy of the selected VHHs in the small intestinal organoids with and without RZ mutations. Selective inhibition of the Wnt signaling pathway is an important issue. The authors provide evidences that the novel VHHs have selective activity against Wnt3/3a stimulation but not to the Wnt1-mediated signaling activity. The main conclusion of the manuscript is strong enough to be considered for publication in Nature Communications.

However, the style of the manuscript still needs further editing. Figure 1 and 2 can be combined into one figure. Figure 4 has very little information, and it can also be combined with Figure 3. In the main text, the description of structural analysis is rather too long and provides too many details. Overall, the manuscript may fit into a short letter format with concise figure sets.

In Figure 6e, it will be better to show Ki67 staining without beta-catenin staining. The green signal of Ki67 is weak to be seen in the merged images.

We thank the reviewers for their enthusiasm for our study and their constructive feedback. We have addressed all points in the point-by-point reply below.

Reviewer #1

1. "What is the advantage of these VHHs over Dkk_C in potential biomedical use? Authors discussed the undesirable lack of Wnt(s)-inhibitory specificity of previously reported LRP6 antibodies. It would be nice to discuss potential issues with human Dkk_C domains, which can bind to the LRP6 P3 domain with nanomolar affinity and should not have any immune problem."

We thank the reviewer for this valuable comment. We now added a discussion of this point to the discussion section, page 9, second paragraph.

2. "Why do all VHHs have two binding affinities with LRP6 P3E3P4E4? What is the implication for Wnt inhibition here? Can authors cross-validate this two-Kd-value phenomenon using other biophysical techniques (e.g. ITC) for the three best-studied VHHs?"

We thank the reviewer for this comment. We now investigated this issue in more depth and we were able to solve the discrepancy. First, we performed the suggested ITC experiments and the results clearly revealed a single binding affinity at nanomolar range, thus matching the crystal structure data. Next, we went back to our QCM approach. We found that the immobilization of LRP6 by a biotin-streptavidin strategy, as shown in the first version of the manuscript, likely caused differences in orientation and accessibility of LRP6^{P3E3P4E4} on the surface, leading to differences in affinities and a 1:2 interaction model. We now repeated the experiments using a different immobilization strategy, in which LRP6 was amine-coupled to the chip, and by performing single cycle kinetic studies (Palau W, Di Primo C. Biochimie. 2012). These results also indicate a 1:1 fitting model for the analyzed VHH. We now replaced the original binding affinity data with the new ITC and QCM results (**new Figure 2c** and **new Supplementary Fig 2**). Results are described on page 4 (last paragraph) and 5 (first paragraph).

3. "Some areas of this manuscript are repetitive."

We now modified the text by removing a number of redundant statements (e.g. statements of RNF43/ZNRF3 deletion leading to Wnt hypersensitivity (p4, top paragraph), and statements on VHH with highest potency (p5, last paragraph)).

4. (minor) "Can authors fix the structure of the Ramachandra Plot outlier residue (Table 1)?"

We fixed the outlier with additional model building and refinement.

5. (minor) "What does the rotation arrow between Figs 3a and 3b mean? Do both Figs 3b and 3d have the same orientation, which is orthogonal with that of 3a?"

Arrow between panels have been deleted as suggested.

Reviewer #2

We thank the reviewer for the appraisal of our study. This reviewer did not raise any comments and recommends publication as is.

Reviewer #3

“The main conclusion of the manuscript is strong enough to be considered for publication in Nature Communications. However, the style of the manuscript still needs further editing. Figure 1 and 2 can be combined into one figure. Figure 4 has very little information, and it can also be combined with Figure 3. In the main text, the description of structural analysis is rather too long and provides too many details.”

We have shortened and rephrased the section on structural analysis to improve readability. We preferred to leave Figure 1 (screen) and 2 (characterization) as separate figures, but we combined Figure 3 and 4 as suggested.

“In Figure 6e, it will be better to show Ki67 staining without beta-catenin staining. The green signal of Ki67 is weak to be seen in the merged images.”

We agree with the reviewer and have adjusted this figure (now Figure 5e) accordingly.

REVIEWERS' COMMENTS:

Reviewer #1 (Remarks to the Author):

Authors have addressed my main concerns.